# VIDES: Video Editing in Seconds with One-Step Diffusion Models

## Abstract

Text-guided video editing with diffusion models is prohibitively slow, hindered by costly multi-step sampling and inversion. We present VIDES, the first framework to successfully adapt one-step text-to-image (T2I) models for high-quality video editing, addressing the core challenges of inversion, editability, and temporal consistency. To bypass slow iterative inversion, we train a learnable encoder that predicts the initial noise for each frame in a single forward pass. This encoder is trained with a novel Structure-Aware Editing (SAE) loss on a curated dataset of structurally-aligned image pairs, teaching it to preserve the source video's geometry during edits. For temporal coherence, we introduce Unified-Frame Editing (UFE), a technique that concatenates frame latents to facilitate cross-frame attention in a single generation step; for long videos, a sliding-window strategy with an anchor frame maintains global consistency. Our extensive experiments demonstrate that VIDES achieves editing quality comparable or superior to state-of-the-art multi-step methods, while operating approximately 155 times faster. This breakthrough paves the way for practical, real-time video editing applications.

## 1 Introduction

Recently, Text-to-Image (T2I) diffusion models (Sohl-Dickstein et al., 2015; Ho et al., 2020; Song et al., 2020b; Rombach et al., 2022) have demonstrated impressive performance, enabling the generation of high-quality, text-aligned images. Building upon these powerful models, the field has rapidly advanced into text-guided image editing. By manipulating the generation process of T2I models, e.g., through inversion techniques (Song et al., 2020a; Mokady et al., 2023) or attention map control (Tumanyan et al., 2023; Hertz et al., 2023), it becomes possible to perform complex edits while maintaining high fidelity to the source image. Motivated by the success of image editing, recent studies have increasingly explored text-guided video editing by leveraging the powerful editing capabilities of T2I diffusion models (Wu et al., 2023; Qu et al., 2024; Cong et al., 2024; Liu et al., 2024; Kara et al., 2024; Wang et al., 2024).

However, unlike image editing, video editing faces a significant challenge: generating video frames is significantly time-consuming due to the multi-step nature of diffusion models. For example, RAVE (Kara et al., 2024), one of the fastest video editing approaches, takes about 24 hours to edit a 5-minute video with 30 FPS [1]. Consequently, despite the growing importance of text-guided video editing in the industry, the high computational cost remains a major obstacle to its practical deployment (Xing et al., 2024).

To overcome the cost issue of video editing, in this paper, we focus on accelerating the image editing process for individual video frames. In general, video editing involves two steps: *inverting* each video frame and *generating* its edited frame conditioned on a modified text prompt. To effectively reduce the time cost of the image generation part, we first adopt one-step diffusion models (Yin et al., 2024a; Dao et al., 2024; Nguyen & Tran, 2024; Yin et al., 2024b; Liu et al., 2023) that have gained attention for their ability to generate images in a single inference step. This approach mitigates the inefficiency of conventional multi-step T2I diffusion models, significantly reducing the time.

However, we find that directly applying one-step diffusion models to the video editing process presents three critical challenges as follows: ***First***, conventional *multi-step inversion methods* (Song et al., 2020a) required for pre-processing are not only *time-consuming* but also yield *blurred and*

---

[1] Measured on a single NVIDIA RTX 6000 Ada GPU.

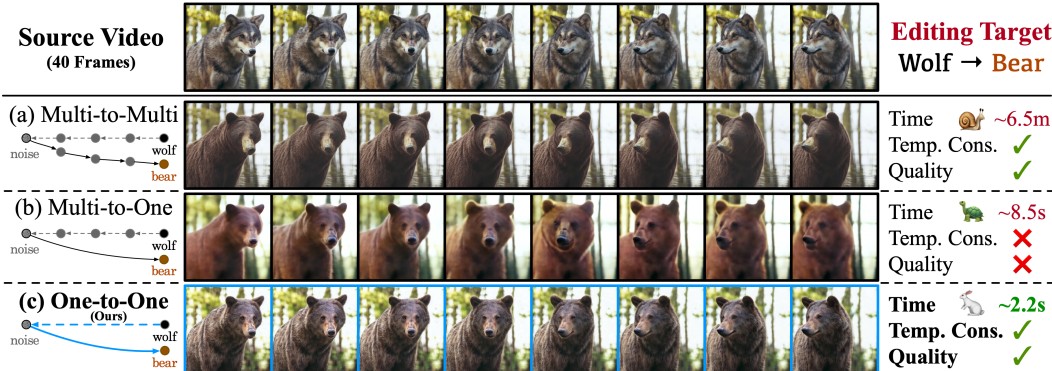

Figure 1: **Three frameworks for text-guided video editing.** *(a) Multi-to-Multi:* The standard approach is high-quality but impractically slow. *(b) Multi-to-One:* A naive adoption of one-step diffusion models fails, producing degraded outputs. *(c) One-to-One (Ours):* Our proposed framework uses a learnable one-step encoder to enable fast, high-quality video editing, successfully resolving the speed-quality trade-off.

*degraded* results when paired with one-step diffusion models (see Figure 1(b)). This degradation occurs because the inversion process, which follows a multi-step trajectory, is a poor approximation for a one-step generator's direct mapping. This fundamental mismatch between the forward and reverse trajectories results in a significant loss of information, washing out fine details. ***Second***, *ensuring structural coherence* during editing with one-step T2I models is difficult. For example, when editing a wolf in an input video to a bear, preserving the wolf's rotating head structure while changing its appearance to the bear is particularly challenging. We empirically find that the straightforward application of existing structure-preserving control methods like Prompt-to-Prompt (P2P) (Hertz et al., 2023) or ControlNet (Zhang & Agrawala, 2023) results in unfaithful edits when constrained to a single inference step. Unlike multi-step processes where guidance is applied gradually to correct errors, one-step models lack this corrective mechanism. Consequently, control signals can overwhelm the generative process, leading to "over-steering" and degraded results. ***Third***, one-step T2I diffusion models do not inherently encode *temporal consistency* information. Consequently, when applied to video editing, they often produce discontinuous or flickering frames (Figure 1(b)). Existing video editing methods based on multi-step T2I diffusion models (Wu et al., 2023; Qu et al., 2024; Cong et al., 2024; Liu et al., 2024; Kara et al., 2024; Wang et al., 2024) address this by finetuning (Wu et al., 2023) or controlling attention maps across different frames (Qu et al., 2024; Cong et al., 2024; Liu et al., 2024; Wang et al., 2024), or by shuffling the denoised noises (Kara et al., 2024) during the multiple sampling steps. However, these multi-step approaches are incompatible with fast video editing, leaving temporal consistency for one-step models an open problem.

To address the aforementioned challenges, we propose three key ideas as follows: ***First***, to overcome substantial processing time and quality degradation in T2I-based video editing, we propose **VIDES** (**VID**eo **E**diting in **S**econds), a pioneering One-to-One video editing framework enabling one-step video inversion and editing. The core of VIDES is a learnable video encoder, a pre-trained U-Net that replaces slow, multi-step inversion with a single forward pass. ***Second***, to mitigate structural collapse in T2I-based video editing, we introduce a novel training approach centered on a Structure-Aware Editing (**SAE**) loss. Instead of relying on external guidance methods that fail in a single step, SAE loss trains the encoder to produce latents that inherently preserve structural integrity even under modified text prompts. By training on a diverse dataset of editing pairs, our encoder learns to disentangle structure from appearance, ensuring that edits are faithful to the source content's geometry, as shown in Figure 1(c). ***Third***, to preserve temporal consistency, we propose Unified-Frame Editing (**UFE**). This method concatenates the inverted latents of consecutive frames into a single, unified latent tensor. When processed by the diffusion model's self-attention layers, this strategy leverages the "emergent correspondence" property (Tang et al., 2023), allowing the model to naturally identify and align features across frames. Moreover, we introduce a sliding window strategy for local coherence and anchor-frame selection for a strong basis. This dual approach effectively ensures both short-term and long-term consistency. Our extensive validation confirms the effectiveness of VIDES, which achieves substantial speed-ups with structural coherence and temporal consistency.

In summary, our main contributions are:

- We introduce VIDES, a pioneering One-to-One video editing framework that, for the first time, successfully adapts one-step T2I models for high-quality video editing. To this end, we design a novel, learnable encoder that replaces slow multi-step inversion with a single, efficient forward pass.

- We propose a Structure-Aware Editing (SAE) loss, a novel training objective that provides the structural preservation capability directly into the inversion process, overcoming the "over-steering" issue common to guidance-based methods in one-step models.

- We present Unified-Frame Editing (UFE), an efficient technique that enforces temporal consistency by processing multiple frame latents simultaneously, leveraging the inherent feature-matching abilities of diffusion models. Paired with a sliding window and an anchor frame, it ensures both short- and long-term coherence.

- Through extensive experiments, we demonstrate that VIDES achieves a remarkable speed-up—approximately 155 times faster than the quickest prior method—while delivering qualitatively comparable or superior results. Our work paves the way for practical, real-time video editing applications.

## 2 RELATED WORK

**Text-Guided Video Editing with Diffusion Models.** To maintain temporal consistency in T2I-based video editing, numerous zero-shot techniques have been proposed, such as fusing attention maps (Qi et al., 2023), propagating token features (Geyer et al., 2023; Wang et al., 2024), using optical flow guidance (Cong et al., 2024), and shuffling noise (Kara et al., 2024). While these approaches have significantly improved video coherence, they all operate within a multi-step sampling framework. This iterative nature, where operations are repeated for each frame across numerous steps, results in prohibitively long runtimes. This computational cost remains a critical barrier, motivating our work on high-quality, single-step video editing.

**One-Step Diffusion Models.** To overcome the significant latency of iterative diffusion sampling, a prominent line of research focuses on distilling multi-step teacher models into highly efficient one- or few-step generators. This paradigm was advanced by methods like Progressive Distillation and Consistency Models, with latent variants (LCM/LCM-LoRA) enabling 2–8 step generation for Stable Diffusion (Salimans & Ho, 2022; Song et al., 2023; Luo et al., 2023). Subsequent works have further refined this approach. Rectified Flow and its successor, InstaFlow, straighten probability-flow trajectories to achieve SD-level quality in a single step via reflow-based distillation (?Liu et al., 2023). Other strategies, such as Adversarial Diffusion Distillation (e.g., SDXL-Turbo, SDXL-Lightning) and various distribution matching techniques (e.g., DMD, SwiftBrush), couple teacher supervision with GAN losses or novel distillation objectives to achieve high-quality synthesis in just 1–4 steps (Sauer et al., 2024; Lin et al., 2024; Yin et al., 2024b;a; Nguyen & Tran, 2024; Dao et al., 2024). These one-step models have revolutionized image generation speed, but this progress has been exclusively image-centric. This creates a critical gap and a major opportunity, as the domain of video editing desperately needs such acceleration. Our work is the first to bridge this gap, introducing the necessary mechanisms to successfully adapt these fast generators for video, thereby unlocking their potential for practical, high-speed applications.

## 3 VIDES: VIDEO EDITING IN SECONDS

In this section, we introduce VIDES, a framework comprising two components—training and inference. Section 3.1 reviews limitations of one-step diffusion–based editing methods and motivates our training design. Section 3.2 addresses temporal consistency in video editing via Unified-Frame Editing (UFE). We show that UFE enables effective cross-frame context sharing, and we propose sliding window and anchor-based schemes to scale the method to long videos. Figure 2 illustrates an overview of our framework.

### 3.1 STRUCTURE-AWARE ONE-STEP INVERSION

A key challenge in adapting one-step diffusion models to video editing is the lack of a generally reliable inversion procedure suitable for one-step diffusion models. In iterative models (e.g.,

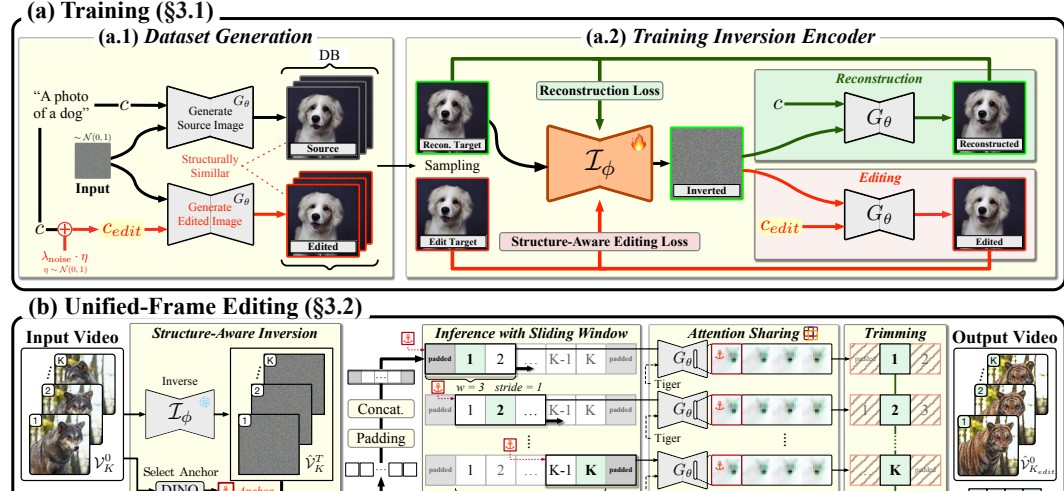

Figure 2: **Overview of our VIDES framework for one-step video editing.** *(a) Training*: We train an inversion encoder using a novel dataset of structurally aligned image pairs generated via *Prompt Perturbation*. The encoder learns to predict an initial noise that supports both faithful reconstruction and structure-aware editing. *(b) Inference*: Our *Unified-Frame Editing (UFE)* method inverts all video frames, concatenates their latents into a single map, and processes them in one pass. This enables cross-frame attention, ensuring temporal consistency. An anchor-based sliding window scales the approach to long videos while maintaining global coherence.

DDPM/DDIM), inversion leverages the multi-step denoising trajectory to recover a latent that faithfully reconstructs a source frame—a cornerstone for identity-preserving video edits. However, one-step models collapse this trajectory into a direct mapping, making standard DDIM-style inversions inapplicable and leaving no widely accepted alternative. This creates a practical bottleneck for faithful reconstruction and temporal consistency in one-step video editing, as illustrated in Figure 1(b).

Therefore, we design and train an inversion encoder $\mathcal{I}_\phi(\cdot)$ for a pre-trained one-step generator $G_\theta(\cdot)$. Given a VAE-encoded latent $z^0$ from a pre-trained autoencoder (Kingma & Welling, 2014; Rombach et al., 2022), prompt embedding $c = E(y)$ for given prompt $y$ (e.g. "a photo of a dog"), and the frozen text encoder $E$ (e.g. CLIP (Radford et al., 2021)), the encoder predicts an initial noise latent $\hat{z}^T = \mathcal{I}_\phi(z^0, c)$. In line with prior inversion methods (Song et al., 2020a; Mokady et al., 2023; Hertz et al., 2023; Tumanyan et al., 2023; Roich et al., 2022; Tov et al., 2021), our objective is to predict a noise latent $\hat{z}^T$ that simultaneously satisfies two criteria: **Reconstruction** and **Editability**.

$$\underbrace{G_\theta\big(\hat{z}^T, c\big) \approx z^0,}_{\textbf{Reconstruction}} \qquad \underbrace{G_\theta\big(\hat{z}^T, c_{\text{edit}}\big) \approx z^0_{\text{edit}}}_{\textbf{Editability}}. \tag{1}$$

While preserving structure preservation is crucial for feasible editability, existing methods like Prompt-to-Prompt (P2P) (Hertz et al., 2023) or ControlNet (Zhang & Agrawala, 2023) are ill-suited for one-step models, as their iterative control logic causes "over-steering" when applied in a single pass (Figure 4). Instead of controlling the generation process, we train an encoder to produce an initial latent $\hat{z}^T$ that inherently encodes the properties required for both faithful reconstruction and structure-aware editing.

To endow the inversion encoder with this structure-aware editability, we proceed in two stages: **1) Dataset Generation with Prompt Perturbation**: We curate a dataset of structurally similar image pairs that share pose and layout but differ visually (Figure 2(a.1)). **2) Training with Structure-Aware Editing Loss**: We train the encoder using a structure-aware editing loss that simulates editing and constrains the output to align with the paired target (Figure 2(a.2)).

**Dataset Generation with Prompt Perturbation.** To generate the required dataset of structurally similar image pairs $DB = \big\{ (z^0, z^0_{\text{edit}}) \big\}$, we propose a simple yet powerful method termed *Prompt Perturbation*. First, we sample a text prompt from a corpus (LAION (Schuhmann et al., 2021), JourneyDB (Sun et al., 2023)) and a random noise vector $z^T \sim \mathcal{N}(0, I)$. The core of our method

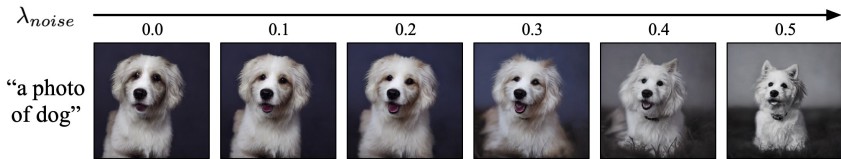

Figure 3: By adjusting the amount of noise injection during image generation, we can create structurally aligned yet slightly different images. These pairs of structurally similar but slightly varied images can then be used as a dataset for our proposed Structure-Aware Editing Loss.

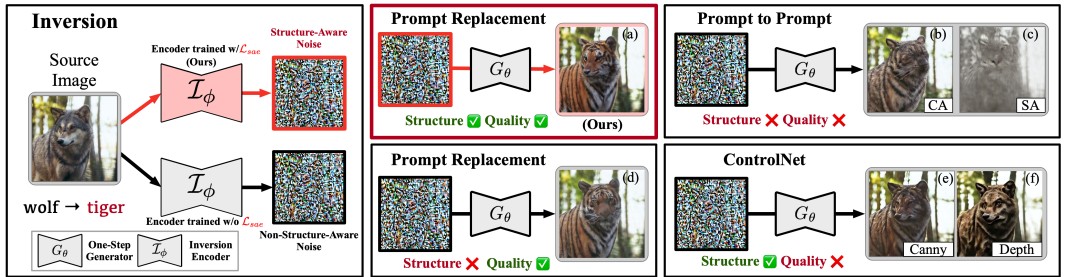

Figure 4: **Comparison of structure-preserving methods for one-step editing.** (a) Our method, trained with the proposed $\mathcal{L}_{\text{sae}}$, maintains structural integrity. (d) Without $\mathcal{L}_{\text{sae}}$, our model fails, leading to structural collapse. (b, c, e, f) Similarly, existing control methods produce heavily degraded results, demonstrating their unsuitability for one-step generation. SA and CA denote the replacement of self-attention and cross-attention, respectively.

is to anchor the generation on the fixed noise $z^T$, ensuring the resulting images share a common structural foundation, while introducing a slight perturbation to the text embedding $c$ to induce a visual change. We formalize this by creating a perturbed embedding:

$$c_{\text{edit}} = c + \lambda_{\text{noise}} \cdot \eta, \quad \eta \sim \mathcal{N}(0, I), \tag{2}$$

where $\lambda_{\text{noise}}$ controls the perturbation intensity (set to 0.1, see Appendix B). Generating from the same $z^T$ with both embeddings yields a perfectly aligned pair for our training objectives (Figure 3):

$$\underbrace{z^0 = G_\theta(z^T, c)}_{\text{Reconstruction Target}}, \quad \underbrace{z^0_{\text{edit}} = G_\theta(z^T, c_{\text{edit}})}_{\text{Editing Target}}. \tag{3}$$

This process allows us to create a vast dataset where each pair shares a structural similarity but differs in appearance, simulating our desired editing behavior.

**Training with Structure-Aware Editing Loss.** Using the generated dataset $DB$, we train the inversion encoder $\mathcal{I}_\phi$ to predict an initial noise $z^T$ that is effective for both reconstruction and editing. Our training objective is formulated using two targets as follows:

$$\mathcal{L}_{\text{total}} = \underbrace{\left\| z^0 - G_\theta(\mathcal{I}_\phi(z^0, c), c) \right\|_2^2}_{\mathcal{L}_{\text{recon}}} + \lambda_{\text{sae}} \underbrace{\left\| z^0_{\text{edit}} - G_\theta(\mathcal{I}_\phi(z^0, c), c_{\text{edit}}) \right\|_2^2}_{\mathcal{L}_{\text{sae}}}, \tag{4}$$

where $\lambda_{\text{sae}}$ is the weight parameter, $z^0 = G_\theta(z^T, c)$ and $z^0_{\text{edit}} = G_\theta(z^T, c_{\text{edit}})$. The first term, $\mathcal{L}_{\text{recon}}$, optimizes the inversion encoder $\mathcal{I}_\phi$ to produce a noise latent $\hat{z}^T = \mathcal{I}_\phi(z^0, c)$ that allows the diffusion model $G_\theta$ to accurately reconstruct the original image $z^0$. The second term, Structure-Aware Editing (SAE) loss, $\mathcal{L}_{\text{sae}}$, ensures that this *same* inverted noise $\hat{z}^T$ also facilitates structure-preserving editing when $G_\theta$ is conditioned on the perturbed text $c_{\text{edit}}$, guiding the edited output to match the pre-generated target $z^0_{\text{edit}}$. Taken together, these terms train $\mathcal{I}_\phi$ to produce an initial noise vector that achieves high-fidelity reconstruction and preserves structural integrity throughout subsequent edits. For implementation, we initialize the inversion encoder $\mathcal{I}_\phi$ with the pre-trained one-step model $G_\theta$, leveraging components from its U-Net. Detailed training settings are provided in Appendix C.

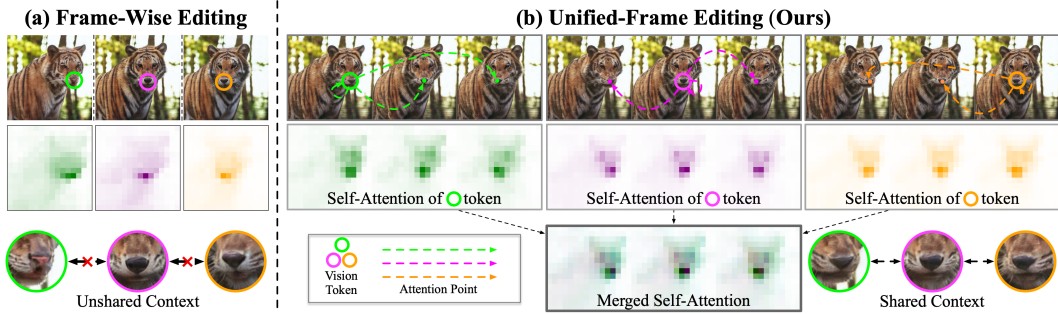

Figure 5: **Mechanism of Unified-Frame Editing for Temporal Consistency.** (a) *Frame-Wise Editing:* Processing frames individually restricts attention to within each frame, preventing information sharing and causing temporal incoherence. (b) *Unified-Frame Editing (UFE):* Our method concatenates all frame latents, allowing the attention mechanism to operate globally. This enables features to be matched and aligned across all frames, enforcing temporal consistency throughout the edit.

## 3.2 UNIFIED-FRAME EDITING

While the inversion encoder from Section 3.1 is effective for single images, applying it to video introduces a new critical challenge: temporal consistency is not preserved. Traditional text-to-image models are designed to operate on individual frames and lack an intrinsic mechanism for handling temporal dependencies. To address this, we propose a novel approach called **Unified-Frame Editing (UFE)**, where inverted latents from multiple frames are concatenated and processed simultaneously in a single inference pass. Overall pipeline is shown in Figure 2(b).

Given a video sequence of $K$ frames, $\mathcal{V}_K^0 = (z_0^0, \ldots, z_{K-1}^0)$, each frame is first inverted using our encoder: $\hat{\mathcal{V}}_K^T = (\hat{z}_0^T, \ldots, \hat{z}_{K-1}^T) = \mathcal{I}_\phi(\mathcal{V}_K^0, c)$. The key challenge is to generate a temporally coherent output. To facilitate information exchange across frames, we spatially concatenate the inverted latents into a single, larger latent map. This is feasible because the U-Net architecture operates on spatial features and can process inputs of arbitrary size (Long et al., 2015; Ronneberger et al., 2015; Rombach et al., 2022). This design allows the generator's inherent self-attention mechanism to function globally, leveraging its ability to align corresponding features across frames (Tang et al., 2023) and thereby enforce temporal consistency. Figure 5 visualizes how UFE fosters this cross-frame communication. Unlike frame-wise editing where information remains isolated, our method allows the attention mechanism to operate globally across the concatenated latents. A token representing a specific feature (e.g., the tiger's nose) in one frame can directly attend to its corresponding feature in other frames. This creates a shared context, ensuring that edits are applied consistently and globally across the entire video sequence.

Let $\hat{z}_k^T \in \mathbb{R}^{C \times H \times W}$ denote the inverted latent for the $k$-th frame. The unified latent map $Z_{\text{UFE}}^T$ is formed by concatenating all $K$ latents along the width dimension:

$$Z_{\text{UFE}}^T = \text{Concat}(\hat{z}_0^T, \hat{z}_1^T, \ldots, \hat{z}_{K-1}^T, \text{axis} = 2).$$

The resulting map $Z_{\text{UFE}}^T$ has dimensions $\mathbb{R}^{C \times H \times (W \cdot K)}$ and is passed to the one-step generator $G_\theta(\cdot)$ with the editing prompt $c_{\text{edit}}$.

**Towards Long Video Editing.** While UFE is effective, its VRAM requirement scales linearly with the number of frames, making it challenging for long videos. To address this, we extend UFE with a sliding window mechanism guided by a global anchor frame. **Sliding Window with Anchor Frame.** Figure 2(b) illustrates the pipeline of our method. The core idea is to process the video in manageable chunks while ensuring both local and global temporal consistency. The sliding window provides local coherence, while the anchor frame enforces a consistent global style and content reference across all windows. First, we identify a representative anchor frame by selecting the medoid in the DINO (Caron et al., 2021) feature space. Its inverted latent, $\hat{z}_A^T$, serves as a global anchor. We validate this choice against other strategies in Appendix D.

We then process the video using an overlapping sliding window approach. For each window of $w$ inverted latents, we prepend the anchor latent $\hat{z}_A^T$ before feeding the concatenated map to the generator $G_\theta$. After generation, the output corresponding to the anchor is discarded, and we extract only the

Table 1: Performance comparison on short (20 frames) and long (90 frames) videos.

| Framework | Method | 20 Frames | | | | | | | 90 Frames | | | | | | |
|---|---|---|---|---|---|---|---|---|---|---|---|---|---|---|---|
| | | SC | BC | MS | AQ | IQ | BQS | FPS | SC | BC | MS | AQ | IQ | BQS | FPS |
| Multi-to-Multi | FLATTEN [‡] | 0.965 | 0.970 | 0.972 | 0.625 | 0.639 | 0.612 | 0.072 | 0.933 | 0.960 | 0.975 | 0.648 | 0.636 | 0.613 | 0.052 |
| | TokenFlow [‡] | 0.983 | 0.976 | 0.991 | 0.668 | 0.680 | 0.663 | 0.075 | **0.974** | **0.975** | **0.990** | 0.622 | 0.668 | 0.632 | 0.084 |
| | FRESCO [‡] | 0.978 | 0.974 | 0.991 | 0.649 | **0.729** | 0.676 | 0.078 | 0.956 | 0.968 | 0.992 | 0.640 | 0.702 | 0.652 | 0.087 |
| | RAVE [‡] | 0.982 | 0.976 | 0.986 | 0.637 | 0.695 | 0.654 | 0.091 | 0.962 | 0.963 | 0.983 | 0.665 | 0.692 | 0.657 | 0.102 |
| | COVE [‡] | 0.983 | 0.976 | 0.989 | 0.645 | 0.655 | 0.639 | 0.061 | 0.955 | 0.964 | 0.988 | 0.651 | 0.650 | 0.630 | 0.041 |
| Multi-to-One | Prompt Replacement [†] | 0.921 | 0.946 | 0.979 | 0.593 | 0.562 | 0.548 | 4.732 | 0.841 | 0.912 | 0.965 | 0.561 | 0.528 | 0.493 | 4.732 |
| | Prompt-to-Prompt [†] | 0.960 | 0.969 | 0.983 | 0.582 | 0.662 | 0.604 | 0.898 | 0.934 | 0.952 | 0.976 | 0.590 | 0.660 | 0.596 | 0.898 |
| | ControlNet (Depth) [†] | 0.968 | 0.961 | 0.984 | 0.658 | 0.673 | 0.646 | 0.578 | 0.945 | 0.944 | 0.978 | 0.626 | 0.652 | 0.611 | 0.578 |
| | ControlNet (Canny) [†] | 0.953 | 0.971 | 0.995 | 0.486 | 0.220 | 0.343 | 0.578 | 0.841 | 0.912 | 0.965 | 0.561 | 0.528 | 0.493 | 0.578 |
| | Plug-and-Play [†] | 0.915 | 0.945 | 0.978 | 0.587 | 0.566 | 0.546 | 0.398 | 0.900 | 0.956 | 0.995 | 0.423 | 0.267 | 0.328 | 0.398 |
| **One-to-One** | **VIDES (Ours)** [†] | **0.983** | **0.977** | **0.991** | **0.678** | 0.703 | **0.679** | **15.625** | 0.958 | 0.965 | 0.989 | **0.670** | **0.723** | **0.676** | **15.793** |

[‡] Multi-Step Diffusion Model (SD1.5). [†] One-Step Diffusion Model (DMD2).

central $s$ frames from the window's output. These segments are then seamlessly stitched together to form the final edited video. This anchor-based sliding window strategy effectively prevents temporal drift by ensuring that each segment is coherent both locally and globally. We set the window size as $w = 7$ and a stride as $s = 5$ through ablation (Appendix E). A detailed, step-by-step formulation of this process is provided in Appendix F.

In summary, VIDES overcomes the three key challenges of one-step video editing: the lack of a *reliable inversion*, *poor structural coherence*, and *temporal inconsistency*. Our one-step inversion encoder, trained with a novel SAE loss, addresses the first two challenges by producing a latent suitable for faithful, structure-aware editing. Building on this, our UFE technique, employing a sliding window and anchor frames, enforces temporal coherence by processing multiple frames jointly in a single pass. This integrated approach unlocks practical, high-quality video editing.

## 4 EXPERIMENTS

### 4.1 EXPERIMENTAL SETUP

We conduct a comprehensive evaluation of our framework across three distinct video editing settings. The first, Multi-to-Multi, encompasses methods that utilize multi-step diffusion models for both video inversion and editing. In this setting, we compare our results against five strong baselines: FLATTEN (Cong et al., 2024), TokenFlow (Qu et al., 2024), FRESCO (Yang et al., 2024), RAVE (Kara et al., 2024), and COVE (Wang et al., 2024). All methods in this category are based on Stable Diffusion 1.5 (SD1.5) (Rombach et al., 2022). The second setting, Multi-to-One, involves a hybrid approach where video inversion is performed via multi-step DDIM, while editing inference is performed in a single step. As no prior work exists for this specific configuration, we construct a baseline by adapting established image editing techniques, including simple Prompt Replacement, Prompt-to-Prompt (Hertz et al., 2023), ControlNet (Zhang & Agrawala, 2023) with depth and canny guidance, and Plug-and-Play (Tumanyan et al., 2023). The final setting is our proposed One-to-One framework, which employs our pre-trained encoder for single-step inversion and performs editing via simple Prompt Replacementment. For both the multi-to-one and one-to-one settings, we use DMD2 (Yin et al., 2024a) as the one-step diffusion backbone. Details in Appendix A.

For evaluation, we curated a dataset of 60 videos from open platforms like Pixabay and prior works, consisting of 51 short videos (20 frames) and 9 long videos (90 frames). We designed five editing prompts for each video—three for local edits (e.g., object change, deletion) and two for global edits (e.g., style change, background modification). This results in 255 video-text pairs for short video evaluation and 45 for long video evaluation. For short videos, the entire 20-frame clip is processed as a single window. For long videos, we employ a sliding window of 7 frames with a stride of 5. All experiments were conducted on a single NVIDIA RTX 6000 ADA GPU.

### 4.2 QUANTITATIVE COMPARISON

We adopt the comprehensive metrics from VBench (Huang et al., 2024) for our quantitative comparison. These metrics are grouped into two main categories: temporal consistency and per-frame quality. For temporal consistency, we measure: Subject Consistency (SC), which evaluates the preser-

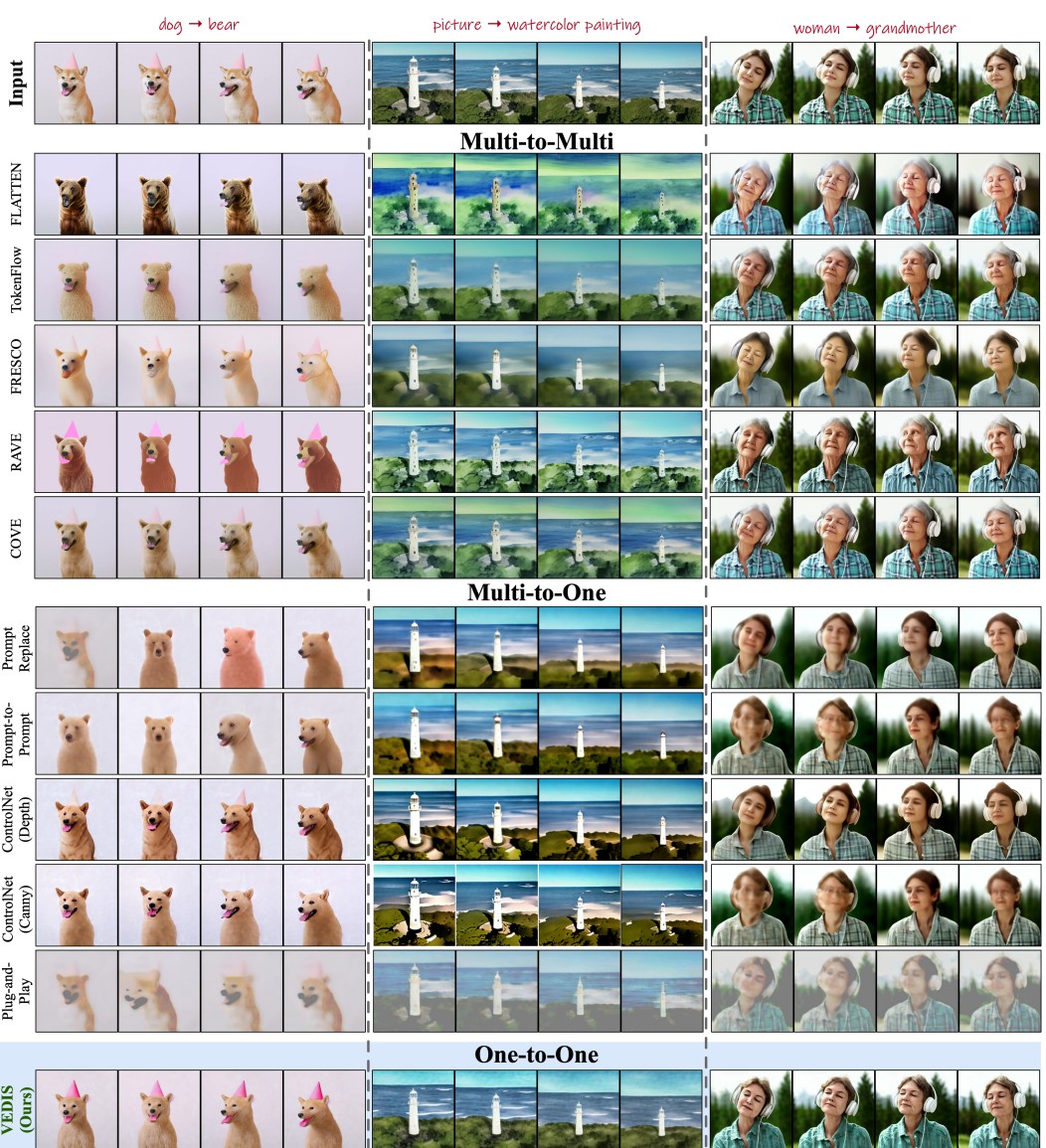

Figure 6: Qualitative comparision.

vation of subject identity using DINO (Caron et al., 2021) feature similarity; Background Consistency (BC), which assesses background stability via CLIP (Radford et al., 2021) features; and Motion Smoothness (MS), which evaluates the fluidity of motion using a pre-trained video interpolation model (Li et al., 2023). For per-frame quality, we measure: Aesthetic Quality (AQ), using the LAION aesthetic predictor (LAION-AI, 2022), and Imaging Quality (IQ), which quantifies perceptual artifacts like noise and blur using MUSIQ (Ke et al., 2021). To provide a single, holistic measure of performance, we introduce the Balanced Quality Score (BQS), defined as: $BQS = \left(\frac{SC+BC+MS}{3}\right) \times \left(\frac{AQ+IQ}{2}\right)$ This multiplicative score ensures a balance between temporal consistency and per-frame quality, penalizing methods that excel in only one dimension. The results are presented in Table 1. Our method, VIDES, achieves state-of-the-art BQS scores for both short and long video editing. Crucially, VIDES is also significantly more efficient, demonstrating speedups of approximately 171× for short videos and 155× for long videos compared to RAVE, the fastest baseline in the Multi-to-Multi framework.

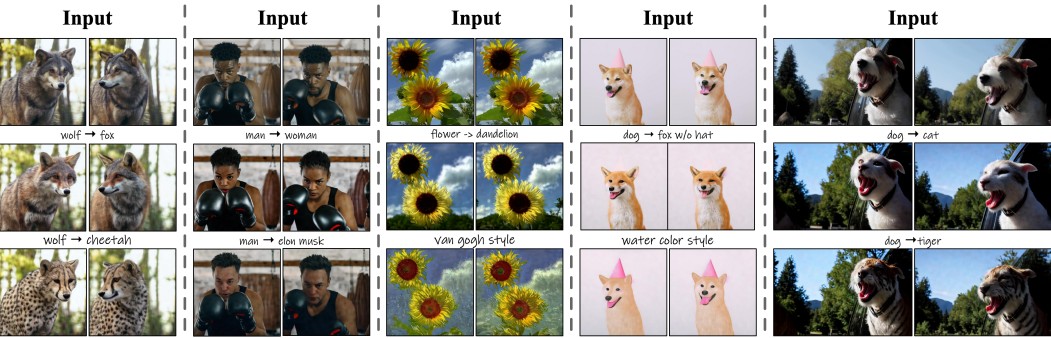

Figure 7: Qualitative results with VIDES.

Table 2: Ablation study on encoder training.

| $\mathcal{L}_{\text{mse}}$ | $\mathcal{L}_{\text{sae}}$ | Structure Distance ($\downarrow$) | CLIP Whole ($\uparrow$) | CLIP Edit ($\uparrow$) |
|---|---|---|---|---|
| $\checkmark$ | – | 0.087 | 21.797 | 19.884 |
| – | $\checkmark$ | 0.074 | **22.349** | 19.863 |
| $\checkmark$ | $\checkmark$ | **0.064** | 22.329 | **20.416** |

Table 3: Ablation study of UFE on 90-frame.

| Sliding window | Anchor | SC | BC | MS | AQ | IQ | BQS |
|---|---|---|---|---|---|---|---|
| – | – | 0.931 | 0.948 | 0.980 | **0.680** | 0.722 | 0.668 |
| $\checkmark$ | – | 0.954 | 0.963 | 0.988 | 0.671 | 0.723 | 0.675 |
| – | $\checkmark$ | 0.943 | 0.950 | 0.985 | 0.672 | 0.722 | 0.669 |
| $\checkmark$ | $\checkmark$ | **0.958** | **0.965** | **0.989** | 0.670 | **0.723** | **0.676** |

### 4.3 QUALITATIVE COMPARISION

Figure 6 provides a qualitative comparison with competing methods. Notably, VIDES successfully modifies objects while preserving their underlying structure. In contrast, other methods used for Multi-to-One setting (row 7-11) often produce results with significant structural collapse or visual degradation. Figure 7 illustrates the versatility of our method across various editing tasks. These include object replacement (columns 1, 3, 5), modifications to human identity (column 2), artistic style transfer (column 3, 4) , and object removal guided solely by a modified prompt (column 4).

### 4.4 ABLATION STUDY

**Ablation Study on Inversion Encoder.** We validate our Structure-Aware Editing Loss (SAEL) in Table 2, evaluating on the PIE-Bench (Ju et al., 2024) using structure distance (Tumanyan et al., 2023) and CLIP scores (Radford et al., 2021). The results demonstrate that while SAEL alone improves structure preservation, it achieves a powerful synergistic effect when combined with the reconstruction loss ($\mathcal{L}_{\text{mse}}$). This combination yields the best overall performance, excelling in both structural integrity and edit accuracy.

**Ablation Study on Unified Frame Editing.** Table 3 presents an ablation study of our Unified-Frame Editing (UFE) method. The results convincingly demonstrate that both the sliding window and anchor-frame strategies significantly improve temporal consistency (SC, BC, MS), exhibiting a strong synergistic effect when combined. We observe a marginal decrease in the AQ score when applying these UFE strategies. While we note a slight dip in the AQ metric with UFE, we consider this an acceptable trade-off for improved temporal stability. This minor artifacting, likely caused by processing the expanded latent representation, is perceptually negligible compared to the stark reduction in flickering and inconsistencies.

### 5 CONCLUSION

We presented **VIDES**, a framework that redefines text-guided video editing by introducing a novel One-to-One pipeline. VIDES unlocks the potential of one-step diffusion models for this task, overcoming the speed limitations of conventional multi-step methods. Our core contributions—a fast, learnable inversion encoder trained with **SAE** loss to preserve structure, and the **UFE** mechanism to enforce temporal coherence—work in concert to resolve the key bottlenecks that previously hindered this approach. Our experiments show VIDES achieves competitive quality and consistency at a fraction of the computational cost, making real-time video editing practical and accessible. Looking forward, we believe the principles of our One-to-One framework could be extended to other video-related tasks, such as video-to-video translation and consistent story generation.

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
