# OpenReview forum: "VIDES: VIDEO EDITING IN SECONDS WITH ONE-STEP DIFFUSION MODELS"
_ICLR.cc/2026/Conference — ICLR 2026 Conference Withdrawn Submission_

### Official Review · Reviewer_6uqy · 2025-10-24

**Soundness:** 3
**Presentation:** 3
**Contribution:** 2
**Rating:** 6
**Confidence:** 4

**Summary:**

This paper presents VIDES, a framework for ultra-fast text-guided video editing using one-step diffusion models. Conventional video editing based on multi-step diffusion suffers from extreme latency (hours for a few minutes of video). VIDES introduces three key innovations to make one-step editing feasible and high-quality:
1. A learnable inversion encoder that predicts the initial noise for each frame in one forward pass, eliminating costly multi-step inversion.
2. A Structure-Aware Editing (SAE) loss trained on structurally aligned image pairs generated by prompt perturbation, ensuring geometry preservation during edits.
3. A Unified-Frame Editing (UFE) mechanism that concatenates frame latents for joint processing, leveraging cross-frame attention for temporal consistency, with a sliding-window and anchor-frame strategy for long videos. Extensive experiments demonstrate a ~155× speedup over prior diffusion-based video editing methods while maintaining comparable or superior visual quality.

**Strengths:**

1. Unified-Frame Editing elegantly leverages global attention for cross-frame consistency.
2. Extensive experiments: both short and long videos, multiple baselines, detailed ablations (SAE loss, UFE, sliding window, anchor frame).
3. Practical scalability — runs on a single GPU and aligns well with industry use cases for real-time editing.

**Weaknesses:**

1. Limited theoretical justification.
The encoder’s success is empirically shown but not theoretically characterized. There is no analysis of how its learned latent space aligns with that of the one-step generator.

2. Dependence on dataset synthesis.
The “prompt perturbation” method for generating structure-aligned pairs is clever but synthetic, potentially limiting generalization to real-world video data.

3. Possible overfitting to static structure.
Since the model is trained on image pairs, it may not fully capture dynamic motion cues or 3D consistency, especially in non-rigid or fast-moving videos.

4. Lack of generalization tests.
Evaluation is limited to short clips (≤90 frames). There is no discussion of performance on videos with complex motion or strong occlusions.

5. Missing resource analysis.
While the paper claims 155× acceleration, a more detailed runtime breakdown (encoder vs. editing, memory footprint) would enhance reproducibility and credibility.

6. Marginal novelty in components.
Each component (encoder inversion, structure-aware training, latent concatenation) builds on existing paradigms, though their joint effectiveness is commendable.

**Questions:**

1. How does VIDES perform on open-domain videos (e.g., YouTube, handheld footage) with uncontrolled motion?
2. Can the SAE loss generalize beyond synthetic pairs to real edit pairs or user-provided source–target pairs?
3. Does concatenating frame latents ever cause spatial artifacts due to large receptive fields near boundaries?
4. How does the learned encoder generalize across different one-step diffusion backbones (e.g., SDXL-Lightning vs. DMD2)?

---

### Official Review · Reviewer_c9a7 · 2025-10-29

**Soundness:** 3
**Presentation:** 3
**Contribution:** 3
**Rating:** 2
**Confidence:** 4

**Summary:**

The authors present VIDES, a fast text-based video editing method that leverages a single step image diffusion model. First, they train a single-step inversion encoder which outputs a noise latent that reconstructs the input image when passed through the diffusion model. A novel prompt augmentation technique is employed during the inversion encoder training to improve the editability of the noise latents during inference. Second, the authors present their Unified-Frame-Editing technique, which allows applying the image editing pipeline for video editing in a temporally consistent manner. It incorporates a sliding window of frames for local edit consistency, and a global anchor frame selection strategy for global edit consistency. The authors provide extensive experiments comparing their method to other image-based video editing methods.

**Strengths:**

1.	The method is fast. It leverages a single step image generator together with a trained single step image inversion model. While the video is processed in a frame-wise manner, batched processing and the single-step inference pipeline make up for it.
2.	The Structure-Aware-Editing (SAE) loss is a nice novelty, enabling the inversion network to output noise maps that are more flexible for editing during inference.
3.	The authors provide several ablations on the design choices for their training pipeline and multi-frame inference strategy.

**Weaknesses:**

Major:

1.	Provided qualitative results are not good. This is my main concern.

   a.	The provided edited results have weak temporal consistency. For example, in the SM files ‘dog_on_car’ edited to tiger - the stripes on the fur change location with each frame. Maybe incorporating some temporal regularization during the inversion process to uncover more correlated noise maps could help with this issue?


   b.	Some of the edited results do not preserve details from the original video well. As seen in the ‘edited_woman_boxing’ edit in the SM files, the background is not consistent with the source video. Maybe localizing the edits using the subject cross attention between text and image tokens could help with maintaining loyalty to the source background?


   c.	The edited results are noisy. As seen in the ‘edited_vangogh style’ example for the watchtower. Are the extracted noise maps following the expected noise statistics for the diffusion model? An evaluation is recommended.


2.	Anchor frame – The editing method selects a single anchor frame to enforce global edit consistency, but for dynamically changing videos it might be problematic. For example, if the scene includes a rotating object the anchor frame will not be able to enforce consistency across all views. Maybe adding several anchor frames such as in [1] [2] could benefit the method. It would be interesting to see the performance of the method on more dynamically changing examples.

Minor:

1.	195 – slightly confusing sentence. “the encoder predicts” – sounds like it references the text encoder from earlier in the sentence.
2.	“preserving structure preservation” – could be phrased differently



[1] Geyer et al. 2024. TokenFlow: Consistent Diffusion Features for Consistent Video Editing

[2] Cohen et al. 2024. Slicedit: Zero-Shot Video Editing With Text-to-Image Diffusion Models Using Spatio-Temporal Slices

**Questions:**

See weaknesses.
Additionally, can the authors provide more video results beyond those 9 included in the supplementary files?

---

### Official Review · Reviewer_Kcgp · 2025-10-31

**Soundness:** 2
**Presentation:** 2
**Contribution:** 2
**Rating:** 2
**Confidence:** 4

**Summary:**

This paper proposes a fast video editing method leveraging one-step text-to-image diffusion models.Starting from the motivation that three bottleneck, (1) time consuming multi-step inversion process, (2) spatial and (3) temporal inconsistency when using T2I model for video editing, the authors propose training inversion network with Structure-aware editing loss and unified frame editing. Qualitative and quantitaive experiments are conducted with various basline with Vbench metric.

**Strengths:**

From the explicit problems in the video editing task, this paper is well motivated and proposes a good research direction that uses T2I model for video editing for efficiency. Training an inversion network with structure-aware editing loss, which uses prompt condition perturbation, is quite novel and impressive. The authors show a lot of quantitative experiments with various baseline comparisons which makes the proposed method confident.

**Weaknesses:**

1. **Poor and Inefficient experiments**
The supplementary visual results reveal substantial flickering and noticeable artifacts throughout the provided videos. The generated videos exhibit clear temporal inconsistency with background distortions, and object-level artifacts are evident. For example, in the “dog on car” → “cat on car” editing example, the cat’s face intermittently appears and disappears at the lower-left corner of the frame.
Such a level of artifact raises serious concerns regarding the robustness of the proposed method and suggests that the model does not perform reliably. In addition, the **temporal flickering score** in the **VBench** evaluation is not reported, further limiting the quantitative assessment of temporal consistency. Moreover, the paper lacks sufficient video comparison results against baseline models, which are necessary for a fair and comprehensive evaluation of visual quality.

2. **Lack of Inference Time Evaluation**
One of the paper’s main claimed contributions is fast video editing. However, there is no quantitative comparison or analysis of inference time against baseline models. Detailed inference time evaluations under varying numbers of frames are also required to substantiate the claimed efficiency.

3. **Proposed Unified-Frame Editing: Not Novel and Questionable for Efficiency**
The proposed unified-frame editing design is not novel. The idea of injecting additional tokens into the attention mechanism is a well-known technique already proposed in several prior works [1,2]. Furthermore, when the video sequence becomes longer, the method may suffer from degraded global consistency due to the limited receptive range of the sliding window.

4. **Insufficient Information on the Inversion Network Training**
The paper does not report sufficient details regarding the training of the inversion network, such as the dataset used or the number of training images. In addition, while the SAE loss introduces a noise weight term $\lambda$, no quantitative analysis or ablation results on this parameter are provided. Such missing details make it difficult to reproduce and verify the claimed improvements.

The paper is well-motivated, and the idea of training an inversion network with SAE loss is novel and interesting. However, the overall qualitative performance is unsatisfactory: the generated videos exhibit **prominent artifacts** and **severe temporal flickering**. Given these significant issues in both robustness and visual quality, I am unable to recommend acceptance of the paper in its current state and must, regrettably, recommend rejection.

---

[1] Geyer, Michal, et al. "Tokenflow: Consistent diffusion features for consistent video editing."

[2] Wu, et al. "Tune-a-video: One-shot tuning of image diffusion models for text-to-video generation."

**Questions:**

1. Why use Gaussian noise for perturbing the prompt condition? I guess there are lots of design choices for perturbing the text condition, such as synonym substitution (e.g, dog-> cat), dropout, and randomly reordering the text.

---

### Official Review · Reviewer_qVB8 · 2025-10-31

**Soundness:** 2
**Presentation:** 2
**Contribution:** 3
**Rating:** 2
**Confidence:** 4

**Summary:**

The paper proposes VIDES, a diffusion-based framework for video editing using a one-step inversion and one-step generation, achieving significant speed improvements over traditional multi-step diffusion-based methods. Specifically, they train a dedicated encoder to produce more faithful inversions that facilitate effective prompt-based edits. After each frame is inverted individually using this encoder, the resulting feature maps are concatenated and processed jointly by a text-to-image (T2I) generator. This design enables the generator’s self-attention mechanism to operate globally across all frames, promoting both spatial and temporal coherence in the edited video.

**Strengths:**

1) The problem the authors tackle—fast video editing—is highly relevant and ambitious. Achieving robustness in this setting could make a substantial contribution to the field of video editing using generative modeling.

2) Training a dedicated encoder within the combined inversion-and-editing paradigm is an interesting conceptual direction that could inspire future research.

**Weaknesses:**

1) Technical justification: The paper’s main novelty—the learnable encoder—is not clearly justified.

a) The motivation for the learnable encoder stems from the authors’ claim that multi-step DDIM inversion faithfully reconstructs source frames, which is inaccurate. Even standard multi-step DDIM inversion (with a finite number of steps, as typically used) cannot perfectly reconstruct images. Moreover, in lines 73–77, the authors argue that one-step inversion is problematic due to the subsequent one-step generator. However, the issue lies not only in the generator but also in the poor inversion quality itself. Overall, the theoretical justification for this argument is weak.

b) The encoder is trained by perturbing the text embeddings, assuming that using the same noise with slightly different prompts produces well-aligned edited images. In practice, this does not hold—small perturbations may preserve alignment but result in negligible edits, while larger perturbations may yield misaligned outputs ([arXiv:2304.06140]). The dataset creation (e.g., wolf → small perturbation) theoretically does not guarantee alignment between source and edited images.

(c) If this encoder works as claimed, it could already enable extremely fast image editing, yet this potential contribution is not emphasized.

2) Lack of quantitative support for runtime: the paper argues that VIDES is ~155× faster than multi-step diffusion-based methods, but no information or quantitative evidence supporting this claim is provided. It is unclear whether this runtime refers to end-to-end editing or only inversion/generation separately. The UNet processes concatenated feature maps (k×width), which can lead to very large self-attention matrices.

3) Experimental validation: only 5 output videos are provided in the supplementary material, which is insufficient to evaluate the method’s performance.

4) Comparisons to baselines: the paper does not compare to modern text-to-video (T2V) models, which are now standard.

5) Presentation and clarity:

a) The paper contains repeated statements, excessive “first, second, third” references, and unnecessary summary at the end of the method section. Equations 1 and 3 are practically identical.

b) Figure 1 (“multi” and “one” part) lacks a clear caption, and the terms are not properly defined when first introduced.

c) Line 342: the use of w and k is confusing; both refer to the number of frames?

d) Line 297: the claim that the UNet works for arbitrary input sizes is not fully supported, particularly for very wide concatenated inputs (k×width), which could introduce artifacts when applied to resolutions or aspect ratios that were not seen during training.

**Questions:**

All my concerns and question were raised in the Weaknesses section

---

### Note · Authors · 2025-11-13

**Comment:**

We sincerely appreciate the time and effort you have dedicated to reviewing our paper. Your detailed and insightful feedback has provided us with valuable perspectives on both the strengths and areas for improvement in our work.

After carefully considering your comments, we recognize that our paper requires significant revisions.

Once again, thank you for your constructive critiques and suggestions. Your feedback has been incredibly helpful in guiding us toward improving our research.

**Withdrawal Confirmation:**

I have read and agree with the venue's withdrawal policy on behalf of myself and my co-authors.